# Evaluation of the Potential of Chitosan Nanoparticles as a Delivery Vehicle for Gentamicin for the Treatment of Osteomyelitis

**DOI:** 10.3390/antibiotics13030208

**Published:** 2024-02-22

**Authors:** Elliot Simpson, Humera Sarwar, Iain Jack, Deborah Lowry

**Affiliations:** School of Pharmacy and Pharmaceutical Sciences, Faculty of Life and Health Sciences, Ulster University, Cromore Rd, Coleraine BT52 1SA, UKsarwar-h1@ulster.ac.uk (H.S.); i.jack@ulster.ac.uk (I.J.)

**Keywords:** nanoparticles, chitosan, osteomyelitis, drug delivery, antimicrobials, gentamicin

## Abstract

Chitosan nanoparticle delivery systems have the potential for enhancing bone healing and addressing osteomyelitis. The objective was to deliver antimicrobial agents capable of preventing or treating osteomyelitis. Each formulation was optimized to achieve desired characteristics in terms of size (ranging from 100 to 400 nm), PDI (less than 0.5), zeta potential (typically negative), and in vitro release profiles for gentamicin. Entrapment percentages varied with gentamicin ranging from 10% to 65%. The chitosan nanoparticles exhibited substantial antimicrobial efficacy, particularly against *P. aeruginosa* and MRSA, with zones of inhibition ranging from 13 to 24 mm and a complete reduction in colony forming units observed between 3 and 24 h. These chitosan nanoparticle formulations loaded with antimicrobials hold promise for addressing orthopedic complications.

## 1. Introduction

Naturally occurring polymers have attracted significant attention in medical science in recent years. This is in part due to their biocompatibility with humans, versatile chemical structures, and environmentally friendly production processes [1]. Among these polymers, chitosan has been extensively investigated because it is readily available in nature, derived from sources like shellfish, insects, and fungi, and it has demonstrated biocompatibility with minimal adverse effects in humans [2,3,4]. Chitosan is the second-most abundant organic polymer after cellulose and has diverse applications in medical research, including wound dressings, tissue engineering, antimicrobial properties, and drug delivery [5,6,7,8,9,10,11].

Chitosan’s unique properties have led to its exploration as a potential carrier for drugs as well as for structural implants to enhance bone healing. Structurally akin to glycosaminoglycan (itself a vital component in collagen formation and bone matrix development), chitosan offers unique advantages in bone tissue applications [12,13]. It has demonstrated osteoinductive properties, attracting bone components like hydroxyapatite and initiating bone regeneration [14]. While most experiments have been conducted in animal models, chitosan holds promise for human applications.

Historically, bone grafts involve harvesting bone tissue either from the patient or a donor (autografts or allografts) for transplantation. Materials like chitosan make ideal substances for synthetic bone grafts [12,13,15]. It would act as a scaffold and provide structural support for bone tissue and reduce the need for pre-surgery scrapings. This, in turn, would lower the risk of complications, such as infection. Moreover, these scaffolds can be enriched with growth factors like BMP 2, 4, and 7 to enhance bone healing [15].

Chitosan-based hydrogels have also been employed as bone implants. These release beneficial agents for bone healing and act as carriers for drug delivery while providing the aforementioned structural support [16,17,18,19]. These hydrogels exhibit malleable properties in terms of shape, size, density, and drug-carrying capabilities. Recent research showcased a chitosan-based hydrogel’s ability to serve as a vehicle for bone mesenchymal cell transplantation. This resulted in reduced inflammation, enhanced vascularization, and tissue knitting in a rat model [18].

Chitosan nanoparticles have been utilized to promote bone healing by delivering growth factors and other therapeutic agents [20]. Chitosan’s ability to encapsulate and release a wide range of payloads, such as antimicrobials, makes it a valuable candidate for bone infections, such as osteomyelitis [21,22,23,24,25,26,27,28,29,30]. The inherent antimicrobial properties of chitosan and its reputation as a reliable drug carrier further support its role as a vector for osteomyelitis treatment [31,32]. In vivo studies using rabbit models have demonstrated that chitosan nanoparticles loaded with vancomycin, when incorporated into chitosan hydrogels, promote bone regeneration and exert antimicrobial effects, offering potential solutions for osteomyelitis treatment [20,33].

The primary objective of this research was to develop an effective method for delivering gentamicin using a chitosan-based nanoparticle delivery system.

## 2. Results

### 2.1. Formulation Optimization

A spectrum of formulations was prepared utilizing varying concentrations of chitosan (ranging from 0.5% *w*/*v* to 2% *w*/*v*) and Tripolyphosphate -TPP (ranging from 0.1% *w*/*v* to 4% *w*/*v*). Following the preliminary optimization, which considered parameters such as size, polydispersity index (PDI), and zeta potential, three formulations were identified (detailed in Table 1). These chosen formulations then underwent a subsequent round of scrutiny involving adjustments to both chitosan and TPP concentrations, with a deliberate focus on concentrations below 1% *w*/*v*. The objective was to ascertain if such alterations could yield a more optimal formulation in terms of the aforementioned parameters.

When the chitosan concentrations were systematically altered while holding the TPP concentration constant at 1% *w*/*v*, an intriguing outcome emerged. All formulations exhibited dimensions falling within the range of 200–391 nm, a range generally regarded as acceptable. However, no discernible pattern or trend could be deduced from this variation. The formulation that exhibited the most promising attributes within this supplementary investigation featured 0.8% *w*/*v* chitosan (with dimensions of 234.20 ± 21.28 nm, a PDI of 0.296 ± 0.02, and a zeta potential of 0.0338 ± 0.14). Nevertheless, it failed to surpass the performance of formulation C6, which consisted of a 1% chitosan to 1% TPP ratio. Consequently, it was deduced that the 1% chitosan: 1% TPP formulation (C6) would be the preferred choice for loading antimicrobial agents.

### 2.2. Effect of Drug Loading on Size, PDI, and Zeta Potential

The selected C6 formulations were loaded with three different concentrations of gentamicin, as mentioned in Table 1, and analyzed for size, PDI, and zeta potential, as shown in Figure 1.

### 2.3. Entrapment of Gentamicin in the Formulation

Entrapment was determined using two methods: direct and indirect entrapment. Both methods of determining entrapment are valid, but it was decided to examine both to facilitate greater accuracy and allow for adjustment if any inconsistencies were reported (Figure 2).

### 2.4. In Vitro Release Studies from Loaded Nanoparticles

The purpose of in vitro testing was to demonstrate the ability of the nanoparticles to release any entrapped agent into a pH-neutral medium (pH 7.4 phosphate buffer saline -PBS), as pH 7–7.4 is the optimum pH for bone resorption and remodeling to occur within the body [34]. Slow controlled release over a full week was ideal, whereas burst release, which is a well-documented issue with chitosan formulations, was considered undesirable [35]. In all cases, sink conditions were maintained, and cumulative release was calculated. Figure 3 shows the cumulative percentage release of gentamicin from the loaded formulations.

#### In Vitro Release Kinetics from Chitosan Nanoparticles

An analysis of the release profiles was conducted to elucidate the specific release kinetics exhibited by each formulation. The particles were scrutinized to discern whether they conformed to zero-order, first-order, or Higuchi release kinetics. The R^2^ values presented in Table 2 were computed based on the release studies conducted for each loaded formulation. Among the formulations generated, the majority exhibited the most congruence with the Higuchi model of release kinetics. Two out of the three formulations demonstrated the Higuchi model as the best-fitting representation for their release profiles, indicative of controlled release from the nanoparticles. One of the formulations displayed first-order release kinetics (CG0.5), signifying that the release was contingent upon both internal and external drug concentrations.

### 2.5. Antimicrobial Effects of Gentamicin-Loaded Chitosan Nanoparticles

The antimicrobial efficacy of the CG formulations was evaluated against relevant bacterial strains to assess their ability to deliver antimicrobial payloads effectively. The assessment of antimicrobial efficacy was conducted through both a zone of inhibition assay and a broth dilution assay.

#### 2.5.1. Zones of Inhibition Produced from Gentamicin-Loaded Chitosan Nanoparticles by Well Diffusion Assay

The loaded nanoparticles exhibited zones of inhibition in which the diameters increased with higher concentrations of loaded gentamicin, as depicted in Figure 4a,b (*p* < 0.001). In contrast, the unloaded C6 nanoparticles failed to produce any inhibition zone against *Pseudomonas aeruginosa* (DSM50071) and methicillin-resistant *Staphylococcus aureus* (ATCC43300), indicating that the chitosan bound within the nanoparticle matrix did not possess inherent antimicrobial effects against both organisms. This observation suggests that any observed inhibition zones were attributed to the drug released from the nanoparticles. The inhibition zones generated by various nanoparticle formulations against *P. aeruginosa* are presented in Figure 4a, while those against MRSA can be observed in Figure 4b.

#### 2.5.2. Broth Dilution Assay

The broth dilution assays conducted for each formulation aimed to illustrate that the suspended nanoparticles could release their antimicrobial payload into a PBS medium over 24 h. Similar to the well diffusion assay, the reduction in colony forming units (CFU) was reliant on both the nanoparticles’ capacity to effectively release an adequate amount of antimicrobial and the susceptibility of the bacterial strain to the specific antimicrobial under consideration. The decline in colony forming units (CFU) of various bacterial strains over time following treatment with gentamicin-loaded formulations is depicted in Figure 5.

### 2.6. Morphological Analysis of Chitosan-NPs

SEM analysis was conducted to explore the morphological structure and aggregation tendencies of Chitosan TPP nanoparticles. The SEM images, as depicted in Figure 6, revealed distinct fine particles alongside small masses of both small and larger particles referred to as agglomerates.

### 2.7. Effect of Gentamicin, Chitosan, TPP, and Blank Formulation on Cell Viability

As anticipated, there was no significant decrease in cell viability observed throughout the 24 h duration in any of the test samples. Figure 7a illustrates that cell viability consistently remained above 95%, indicating minimal impact on cell viability during the 24 h exposure to these agents. The concentrations used align with the anticipated release profile from the particles during the initial 24 h period, suggesting the safety of each agent encapsulated within the nanoparticles for use in this specific cell line. Consequently, these findings affirm the likelihood of the safety of these agents for application in bone tissues, particularly in the treatment of osteomyelitis. The impact of the constituents of chitosan nanoparticles on SAOS-2 cell viability is depicted in Figure 7b. Analogous to the behaviour observed with the active agents, both the components of the chitosan nanoparticles (chitosan and TPP) and the C6 formulation exhibited minimal effects on cell viability. This consistent pattern implies a low risk associated with the use of these nanoparticles in the SAOS-2 cell line. The non-toxic characteristics of the nanoparticle components, coupled with the marginal influence on cell viability demonstrated by the C6 nanoparticle formulation, suggest that this specific formulation is likely safe for application in the SAOS-2 cell line and other bone tissues. This supports the potential use of the C6 formulation for enhancing bone healing and treating osteomyelitis.

### 2.8. Statistical Analysis

The statistical analyses aimed to elucidate the influence of diverse drug loading concentrations on key formulation parameters. Notably, the release of gentamicin exhibited a significant difference in means across concentrations (0.5%, 1%, and 2%), as evidenced by the ANOVA (*F* = 17.00, *p* < 0.001). This suggests a substantial impact on the release profile, underscoring the formulation’s sensitivity to varying drug concentrations. Conversely, the ANOVA results for particle size showed no statistically significant differences (*F* = 0.73, *p* = 0.50) among loading concentrations (0.5%, 1%, and 2%), but the comparison of blank formulation with loaded formulations showed statistically significant results (*F* = 31.00, *p* ≤ 0.0001). PDI analysis hinted at a potential influence of drug loading concentrations (0.50%, 1%, and 2%) on particle dispersion characteristics; although, the result approached but did not reach statistical significance (*F* = 3.66, *p* = 0.051). In contrast, zeta potential analysis revealed no significant effect of loading concentrations (0.50%, 1%, and 2%) on particle charge (*F* = 0.11, *p* = 0.90). The adoption of ANOVA was prudent, given its appropriateness for comparing means across multiple groups. These statistical findings provide valuable insights for optimizing the formulation, ensuring controlled drug release, particle stability, and desirable characteristics at varying drug loading concentrations.

## 3. Discussion

Chitosan:TPP nanoparticles were formulated by adding TPP to a chitosan solution, inducing crosslinking and nanoparticle formation through stirring. This process incorporates agents into the chitosan:TPP matrix. These nanoparticles offer advantages in drug delivery and bone healing. Chitosan is biocompatible, biodegradable, promotes osseointegration, and can deliver drugs to bone tissue. While used in bone applications, limited data exist on free nanoparticles in bone tissue. Desired loaded nanoparticle parameters include size 100–400 nm, PDI < 0.5, and negative zeta potential (−20 to −40 mV). Unloaded particles should meet these criteria but be closer to 100 nm in size due to agent entrapment [35,36,37].

Variations in TPP and chitosan concentrations significantly impact the size and stability of the resulting particles. This influence extends beyond the particle composition and extends to the rate of the crosslinking reaction, a factor that has been shown to substantially modify particle properties [38]. Formulations created with 0.5% and 1% *w*/*v* chitosan (C1–C8) exhibited sizes below 600 nm (ranging from 120 to 560 nm) and maintained acceptable PDI values (ranging from 0.2 to 0.46), consistent with prior research findings [39]. However, formulations stemming from 1.5% and 2% *w*/*v* chitosan (C9–C16) displayed considerable fluctuations in both size and PDI values. Previous studies have underscored the strong influence of chitosan concentration on these parameters [35,36], emphasizing that extreme concentrations of both chitosan and TPP can yield variable outcomes [40,41].

Most zeta potentials were positive but are so close to zero that they could be considered neutral, and this is also true in the case of the negative zeta potentials. It would normally be expected that, with variation of TPP and chitosan, the zeta potential would become more positive or negative depending on which agent took predominance on the surface of the particle, as has been demonstrated in various other studies [41,42].

Unloaded nanoparticles comprised chitosan polymer and TPP crosslinker in varying ratios to one another. As has been demonstrated, varying these ratios will lead to varying effects on the particle size, PDI, and zeta potential. When loaded with an active or model agent, those agents will influence the characteristics of the particles. Size is the characteristic with the most predictable change that occurs after entrapment [43]. Typically, when a drug is loaded, sterically it will be contained within the matrix, and it will cause the diameter of the particle to expand. Changes to zeta potential can also be predicted based on the charge held by the agent being loaded, i.e., a negatively charged agent will lead to a more negative zeta potential [44]. PDI is more difficult to predict, as various factors can affect PDI, such as surface area, free energy within the system, and forces of attraction/repulsion between particles.

Gentamicin loading was anticipated to increase particle size, as the drug’s incorporation into the chitosan:TPP matrix displaced space and caused swelling, resulting in larger particles than their unloaded counterparts. It is reasonable to expect a more significant size increase with higher drug loading. The effect on the polydispersity index (PDI) depends on the interaction between the entrapped drug and nanoparticle matrix, making it challenging to predict. Zeta potential was expected to correlate with the charge of the entrapped drug, affecting surface charge. An increase in loading concentration led to more drug within the particles. Gentamicin presence was expected to generate a positive zeta potential, and there was indeed a significant increase in size with increasing gentamicin loading (*p* < 0.001). PDI values met the predetermined criteria, staying below 0.5.

Figure 2 illustrates that gentamicin resulted in lower zeta potentials than the unloaded nanoparticles. This contradicts the expected outcome, where gentamicin would increase zeta potentials due to its positive charge and presence within the nanoparticle matrix, as demonstrated in previous work [35,45]. This discrepancy could be attributed to interactions between gentamicin and TPP, neutralizing surface charge as the polycationic gentamicin may attract more anionic TPP. There was no significant difference in zeta potential with increasing gentamicin concentration.

As the added drug amount increases, drug content within nanoparticles generally rises, while entrapment efficiency tends to decrease or stabilize [9]. Each formulation has an optimal drug capacity, and exceeding it hinders further drug incorporation, reducing entrapment efficiency [46]. Indirect entrapment values were chosen for future percentage release calculations, and direct entrapment was examined for validation. Nevertheless, the particles seem to have incorporated a sufficient amount of the drug to exhibit antimicrobial effects [47]. It was expected that the formulations with the highest drug loading would release the most drug, as these formulations would have more drug present within their structure. Consequently, these formulations should be capable of maintaining a concentration gradient in sink conditions for longer, as the depot of drug within the particle will take longer to deplete.

The percentage release from the particles is displayed in Figure 3. The in vitro cumulative release from gentamicin-loaded formulations aligns with expectations: that the cumulative drug amount released over seven days significantly increases with loading concentration for CG0.5, 1, and 2 (*p* < 0.0001). Gentamicin release profiles exhibited rapid initial increases in concentration over the first day (Figure 3), followed by continuous gradual release over the subsequent seven days. In CG2-loaded particles, competition for drug exit via any developing pores may explain the release pattern. CG1 and CG2 exhibited a higher gentamicin percentage release than CG0.5. CG0.5 and CG1 release around 90% of their payload over seven days, while CG2 releases 57.44 ± 2.72%. Previous studies attempting to control gentamicin release achieved 70% release over 8 h, with complete release by 24 h [48].

Ji et al. (2011) found that chitosan:TPP nanoparticles loaded with salicylic acid and gentamicin released 45–85% of gentamicin in the first 10 h, followed by an additional 10% release up to 70 h, with a maximum release of 55–90%. These release levels are comparable to the CG formulations, showing similar release after 24 h [45].

Disk diffusion assays use small disks (5–10 mm) loaded with antimicrobials to demonstrate the ability of an agent to diffuse into the surrounding agar tissue and prevent the growth of an inoculated microbe around the disk. The current study was carried out by adding the formulation to a well in the agar (10 mm) [49]. The zones produced from the different nanoparticle formulations against *P. aeruginosa* are displayed in Figure 4a. Each loading concentration produced a sequentially larger zone (G0.5: 18 ± 0.82 mm, CG1: 21 ± 1.41 mm, and CG2: 24 ± 0.82 mm), and they were significantly different from one another (*p* < 0.005). These were comparable to a concentration of free gentamicin solution (0.5 mg/mL: 16.33 ± 0.47 mm, and 1 mg/mL: 20.33 ± 0.47 mm). These results show that, over the 24 h incubation period, the particles released enough gentamicin to cause antimicrobial effects, and also based on other studies, they suggest that the strain is susceptible to gentamicin [50]. The zones generated against MRSA can be seen in Figure 4b. Against-MRSA zones for both nanoparticles and standards were smaller than those generated against *P. aeruginosa* with an increase in loading concentration, and the zone of inhibition increased in diameter (CG0.5: 13.17 ± 0.47 mm, CG1: 17.33 ± 0.94 mm, and CG2: 18.17 ± 0.85 mm). In this instance, the CG0.5 loaded particles were more comparable to the 0.25 mg/mL standard (14 ± 0.82 mm), and when error is considered, both the CG1 and CG2 formulations were comparable to the 1 mg/mL gentamicin standard (17.33 ± 0.24 mm). These results show that MRSA is more resistant to gentamicin than *P. aeruginosa* and is comparable to results demonstrated in previous work [51].

Based on the results from the release study and zone of inhibition assay generated from the gentamicin-loaded particles, the results of the broth dilution assay were expected to resemble similar trends that, with the increasing concentration of gentamicin loading, there would be a more rapid reduction in colony forming units over time. *P. aeruginosa* and *S. aureus* are both known to be susceptible to gentamicin. P. aeruginosa DSM50071 is susceptible to gentamicin; however, MRSA ATC43300 has been demonstrated to be resistant to gentamicin at lower concentrations. Because of these differences in the strains, it was expected that MRSA would take longer to show an antimicrobial effect (if any at all) for each formulation when compared to *P. aeruginosa*.

There was no observed reduction in microbial growth in the control sample. The results for *P. aeruginosa* (as displayed in Figure 5a) were as expected; the CG2 formulation caused a total reduction in CFU by 3 h, followed by the CG1 formulation at 4 h, with the CG0.5 formulation taking up to eight hours to show the total reduction in growth. The results concur with what was initially predicted, as well as corroborating both the release and zone of inhibition results, that increasing loading concentration will lead to an increase in observed activity against P. aeruginosa. Other strains of *pseudomonas* have shown similar times taken to show the complete reduction of CFU, taking between 3 and 24 h to show complete eradication [52]. As with the previous bacterial strains, no reduction in bacterial growth was observed in the control samples. As predicted, MRSA required more gentamicin to be released from the particles to elicit the same response as seen in *P. aeruginosa*. As the results in Figure 5b show, the full 24 h were required to reduce microbial growth to zero for all formulations. However, it can also be seen that there was a reduction in CFU starting in all formulations, beginning from the 3 h time point, with each formulation showing a gradual reduction in CFU until each formulation showed no growth at 24 h. These results are similar to what was observed in a paper examining the pharmacodynamics of gentamicin against *S. aureus* using different concentrations of gentamicin, in which it was observed that there was a gradual reduction in Log CFU/mL over 24 h; however, they did not observe a complete reduction in Log CFU/mL [34].

The MTT assay was utilized to assess the impact of the agents on the SAOS-2 cell line. Each loading agent, along with a sample of blank nanoparticles, underwent an evaluation to determine whether the particles or their loading agents induced any adverse effects on cell viability, thereby influencing their suitability for further cell-related investigations. The safety of nanoparticles in the SAOS-2 cell line is crucial, especially concerning their potential as a platform for BMP-2 loading. Given the prior use of chitosan in human tissues, including bone tissues [53], cytotoxic effects were not expected. The significance of this assay lies in its ability to establish the safety of particles and their payloads in bone tissue. Any evident cytotoxic effects would render the formulation or agent unsuitable for contributing to bone tissue healing or the treatment of osteomyelitis. The data obtained from the present study support the potential use of the C6 formulation for enhancing bone healing and treating osteomyelitis without altering the cell viability.

## 4. Materials and Methods

### 4.1. Materials

Medium molecular weight chitosan (99%), trifluroacetic acid (99%), methanol (99%), gentamicin sulfate (99%), acetic acid, KCl (99%), HCl (99.9%), dialysis tubing with a molecular weight cut off of 14,000 Da, tryptic soy broth and agar, and PBS tablets were all purchased from sigma Aldrich. All bacterial strains were grown from lawns already kept by the microbiology department. The strains used were *P. aeruginosa* DSM50071 and MRSA ATC43300.

### 4.2. Methods

#### 4.2.1. Synthesis of Chitosan Nanoparticles

The 1% *w*/*v* chitosan was dissolved in 5% *w*/*v* acetic acid at 50 °C with continuous stirring until a homogenous solution was obtained. The crosslinker TPP was dissolved in deionized water (dH_2_0) to obtain a concentration of 1% *w*/*v*. If a drug was to be loaded, an amount of powder was added to the chitosan solution to achieve a loading concentration (Table 3) under continuous stirring with a magnetic stirrer until the powder was dissolved. TPP solution was added dropwise into corresponding chitosan solutions with continuous stirring. The solution was stirred for an hour to ensure that the crosslinking reaction was completed before being centrifuged at 8000 rpm at room temperature for an hour. Any pellets formed were washed with dH_2_0, and the centrifugation process was repeated. Pellets (and supernatants if drugs were being loaded) were collected for analysis.

#### 4.2.2. In Vitro Detection of Gentamicin

The release of gentamicin from chitosan nanoparticles was performed using a dialysis method. Briefly, the pellet was resuspended in 10 mL of PBS, and 1 mL of this was placed in a cellulose dialysis membrane. This membrane was then immersed in 9 mL of PBS and placed into a shaking incubator at a temperature of 37 °C and a speed of 100 rpm. Samples of 1 mL were then taken at pre-determined time intervals for analysis. Sink conditions were maintained by replacing samples taken with an equal volume of PBS. The samples were analyzed using HPLC.

The detection of gentamicin was assessed by reversed-phase HPLC using a Waters-Breeze system with a UV detector set to 254 nm. Reverse-phase HPLC was performed using a Phenomenex Luna C18 column 100A (150 × 4.6 mm) with a 5 μm particle size at room temperature, as a stationary phase, and a 10:90 methanol: 0.1% *v/v* Tetrafluroacetic acid (TFA) in dH_2_0, as a mobile phase, with a flow rate of 1 mL/min. The injection volume was 150 μL. Standards were injected before and after each set of samples consisting of 0.2, 0.4, 0.6, 0.8, and 1 mg/mL of gentamicin to demonstrate the consistency of readings.

#### 4.2.3. Chitosan Nanoparticle Entrapment

Pellets and supernatants were both analyzed to determine direct and indirect entrapment, respectively.

##### Indirect Entrapment

Supernatants were analyzed using an appropriate method for the drug being examined. When examining nanoparticles containing gentamicin, HPLC was used as described in Section 4.2.2 to determine the concentration in the supernatant. This was then extrapolated to determine the amount of drug present in the supernatant. The amount of drug present in the supernatant was subtracted from the total amount of drug added initially; this indicated how much drug was left in the pellet, and from this value, an entrapment efficiency was calculated using Equation (1).
(1)$$EE\;\left(\%\right)=\left(\frac{Total\;amount\;of\;drug\;added-Amount\;of\;drug\;in\;supernatant}{Total\;amount\;of\;drug\;added}\right)\times 100$$

##### Direct Entrapment

Before analysis of a pellet could be performed, it was destroyed by placing the pellet in a solution of 0.1 M HCL under continuous stirring for 24 h at room temperature. The resulting solution was then centrifuged at 8000 rpm, and the supernatant was collected for analysis using the methods described previously for each drug. The entrapment efficiency was calculated using Equation (2).
(2)$$EE\;\left(\%\right)=\left(\frac{Amount\;of\;drug\;in\;Pellet}{Total\;amount\;of\;drug\;added}\right)\times 100$$

#### 4.2.4. Antimicrobial Studies

##### Plating of Media

Tryptic Soy Agar (TSA) was used to culture the two aerobic strains *P. aeruginosa* and Methicillin-resistant *S. aureus*. The media were prepared according to the manufacturer’s guidelines by dissolving in the appropriate solvent and then autoclaving to ensure that the media were sterile. Approximately 20 mL of TSA was transferred aseptically to an agar plate and allowed to cool and solidify. The plates were then placed in a cold room until required.

##### Growth of Bacterial Cultures

Aerobic bacteria (*P. aeruginosa* and Methicillin-resistant *S. aureus*) were grown in Tryptic Soy Broth (TSB) overnight in an incubator at 37 °C to a 0.5 McFarland standard. They were then streaked onto TSA plates using sterile swabs, incubated overnight, and stored in a cold room until they were required for analysis.

##### Zone of Inhibition Control Assay

As previously described, a broth was produced and inoculated with a strain of bacteria at 0.5 McFarland standard. A total of 100 μL of the broth was then placed onto an agar plate and, using an L-shaped spreader, was evenly distributed across the surface of the agar. Using a sterile 1 mL pipette tip, wells were dug into the agar in a pattern displayed in Figure 8. In each well, 50 μL of the sample was placed, allowing each plate to hold triplicates for each sample set being investigated. A range of standards listed in Table 4 were plated depending on which strain was being investigated. Control plates containing unloaded nanoparticles and empty wells were also plated. Plates were incubated in a manner appropriate to the strain being investigated overnight. After incubation, a ruler was used to measure any observed zone of inhibition. In the case of an irregularly shaped zone, the most representative diameter was taken.

##### Determination of Antimicrobial Activity of Antimicrobial Loaded Nanoparticles

The antimicrobial activity of nanoparticles loaded with gentamicin was evaluated using a broth dilution method. A strain of bacteria was grown, as previously outlined, to a McFarland standard of 0.5 in an appropriate broth. This resulting solution was then diluted 1:100 with sterile PBS. A total of 9 mL of this resulting suspension was then transferred aseptically to a bijou vial. The chitosan nanoparticle pellet being examined was resuspended in 10 mL of PBS, and 1 mL of this was transferred to the bijou vial. A bijou vial was prepared in this manner for each nanoparticle formulation being examined and for controls, and an additional setup was prepared for each sample at 24 h. The vials were placed in an appropriate incubator and incubated at 37 °C. Samples of 100 μL were taken at hourly intervals for the first eight hours and then at an additional time point at 24 h. Sample volume replacement was not required, as the volume being removed was too small to affect growth, as previous work has demonstrated. Each sample taken was further serially diluted from a range of 10^−1^ to 10^−6^, plated as displayed in Figure 8b, and incubated overnight at 37 °C. Any colonies present on the plates were counted and, using equation 3, colony forming units (CFU) were determined.
(3)CFU=Average colony count×50×Dilution factor

##### Scanning Electron Microscope (SEM) Analysis

Sample dried powder preparations were mounted on double-sided adhesive tape and placed on aluminum stubs. Before imaging under SEM, the samples were coated with 10 nm gold in a Quorum Q150 sputter coater (Quorum Technology Ltd., Lewes, UK) to allow electrical conductivity then were imaged with SEM (Steroscan 90, Cambridge Instrument, London, UK).

##### Cell Culture and Toxicity

SAOS-2 cells, sourced from ATCC, were cultured in McCoy’s 5a Modified medium supplemented with 15% fetal bovine serum and 1% penicillin-streptomycin. The cells were maintained in a controlled environment at 37 °C with 5% CO_2_. Media changes occurred every 3–4 days, and cell passaging took place 6–7 days post-seeding when cells reached 70–80% confluency. The process involved discarding spent media, rinsing the flask with sterile PBS, and treating the cells with trypsin/EDTA solution for 5–10 min to detach them from the flask wall. Visual confirmation of cell detachment was achieved under a light microscope. The trypsin was neutralized with fresh media, and cells were centrifuged at 1000 rpm for 10 min. The supernatant was discarded, and the cell pellet was resuspended in pre-warmed media, followed by splitting into two flasks.

For cryopreservation, the same splitting process was followed, but instead of pre-warmed media, a cryopreservative solution (McCoy’s 5a Modified medium with 5% DMSO) was used after centrifugation. One milliliter of the resulting suspension was aliquoted into cryovials, frozen at −80 °C, and then transferred to liquid nitrogen for long-term storage.

##### In Vitro Cell Toxicology Assay

Before investigating the potential of nanoparticles to enhance bone healing, it is imperative to establish the non-toxic nature of both the particles and their loading agents on the SAOS-2 cell line. SAOS-2 cells were cultured and trypsinized according to the procedures outlined above. In preparation for the toxicity assay, cells were adhered to a 96-well plate. This was accomplished by resuspending the cell pellet, obtained after centrifugation, in fresh media to achieve a concentration of 75,000–100,000 cells/mL. Subsequently, 100 µL of this cell suspension was added to specific wells in a 96-well plate and incubated overnight at 37 °C with 5% CO_2_.

Samples of serum-free media were then supplemented with nanoparticle components, metronidazole, gentamicin, and BSA. These modified media samples were introduced into the respective wells and further incubated for 24 h. Following this incubation period, 20 µL of 5 mg/mL thiazolyl blue tetrazolium bromide (MTT) was added to the designated wells, along with some unseeded wells to serve as controls. The plate underwent a subsequent 3.5 h incubation at 37 °C. After aspirating the wells, 150 µL of DMSO was added to the relevant wells. The plates were shielded from UV exposure with tin foil and agitated using an orbital shaker for 20 min. Finally, the plate was read at 590 nm using a microplate spectrophotometer, and the optical density of the wells was recorded. Percentage cell viability was calculated using Equation (4).
(4)$$Cell\;Viability\;\left(\%\right)=\frac{\left(OD\;of\;treated\;well-OD\;of\;Control\right)}{\left(OD\;of\;untreated\;well-OD\;of\;blank\right)}\times 100$$

## 5. Conclusions

This study successfully formulated stable chitosan and TPP particles capable of loading various concentrations of gentamicin. Each formulation was optimized to achieve the desired characteristics in terms of size (ranging from 100 to 400 nm), PDI (less than 0.5), and negative zeta potentials. Entrapment percentages varied with gentamicin ranging from 10% to 65%. The safety of these particles and their loaded agents was affirmed through SaOs2 cell line testing, showing a cell viability of ≥95%. In vitro antimicrobial release studies demonstrated effectiveness, with up to 90% release over 7 days, achieving significant bacterial reduction within 3 h for the formulation with the highest drug concentration. Antimicrobial efficacy was demonstrated, particularly against *P. aeruginosa* and MRSA, with zones of inhibition ranging from 13 to 24 mm and a complete reduction in colony forming units observed between 3 to 24 h. Comparative efficacy analysis indicated promising results compared to existing formulations. These findings suggest potential applications in enhancing bone healing and preventing/treating infections, either through incorporation into scaffolds or hydrogels or as standalone treatments. Further research, particularly in vivo studies, is warranted to validate and extend these promising results.

## Figures and Tables

**Figure 1 antibiotics-13-00208-f001:**
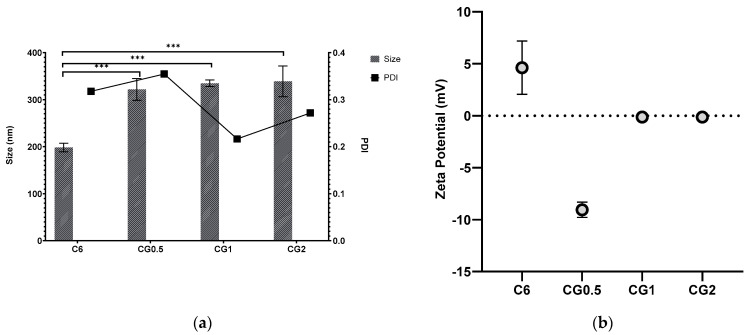
Particle size, PDI (**a**), and zeta potential (**b**) of formulation C6 chitosan: TPP nanoparticles loaded with different concentrations of gentamicin (*** *p* < 0.001).

**Figure 2 antibiotics-13-00208-f002:**
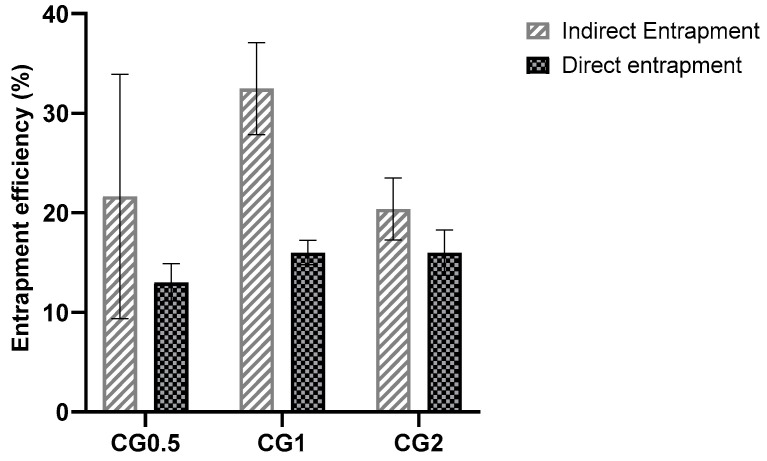
Direct and indirect entrapment efficiencies of C6 chitosan: TPP nanoparticles loaded with different concentrations of gentamicin.

**Figure 3 antibiotics-13-00208-f003:**
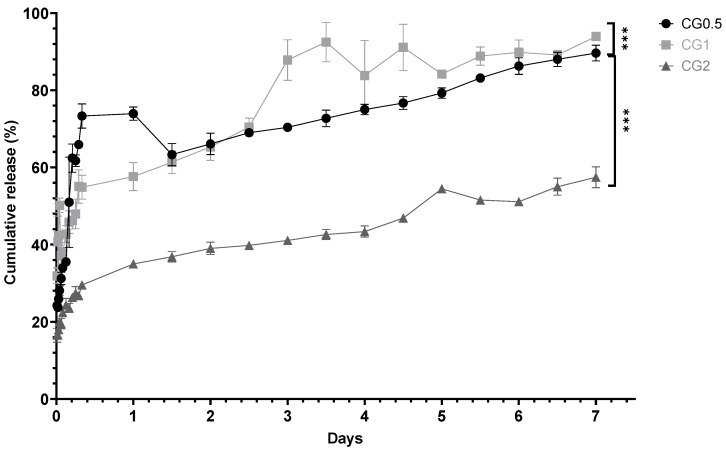
In vitro cumulative percentage release of gentamicin from chitosan nanoparticles loaded with various concentrations of gentamicin over seven days (*n* = 3) (*** *p* < 0.001).

**Figure 4 antibiotics-13-00208-f004:**
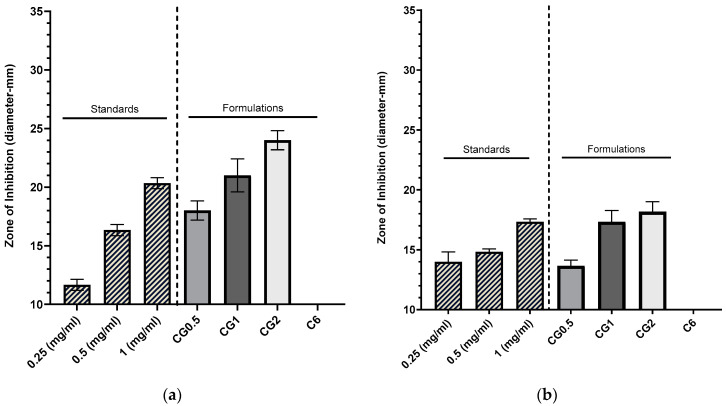
Zones of inhibition produced using CG nanoparticles loaded with various concentrations of gentamicin, gentamicin standards, and a C6 nanoparticle control against *P. aeruginosa* DSM50071 (*n* = 3) (**a**). Zones of inhibition produced using CG nanoparticles loaded with various concentrations, gentamicin standards, and a C6 nanoparticle control against methicillin-resistant *S. aureus* ATC43300 (*n* = 3) (**b**).

**Figure 5 antibiotics-13-00208-f005:**
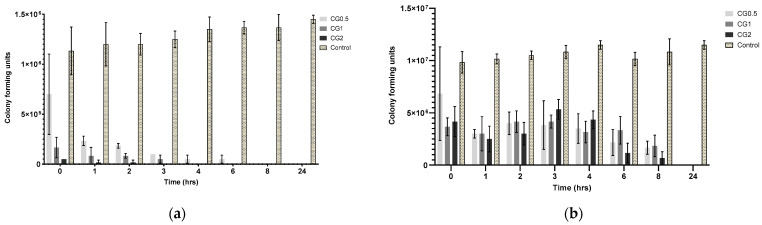
Colony forming units (CFU) recorded for: (**a**) *P. aeruginosa* DSM50071 when subjected to CG nanoparticles loaded with different gentamicin concentrations and a C6 nanoparticle control over 24 h (*n* = 3); (**b**) MRSA ATC43300 in response to CG nanoparticles loaded with various gentamicin concentrations and a C6 nanoparticle control over 24 h (*n* = 3).

**Figure 6 antibiotics-13-00208-f006:**
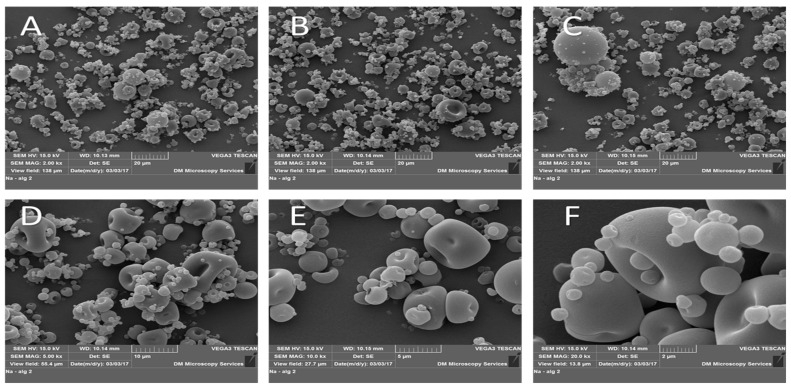
SEM images of chitosan-TPP NPs. (**A**–**C**): 2000× magnification, (**D**): 5000× magnification, (**E**): 10,000× magnification, and (**F**): 20,000× magnification. Sample powders were mounted on double-sided adhesive tape and placed on aluminium stubs; samples were coated with 10 nm gold in a Quorum Q150 sputter coater, then imaged with SEM with STEREOSCAN 90.

**Figure 7 antibiotics-13-00208-f007:**
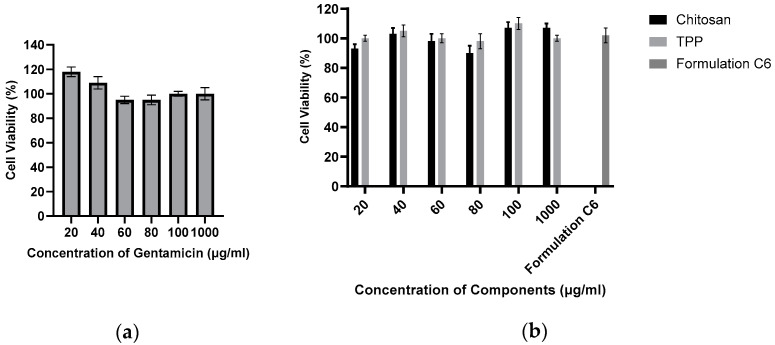
Percentage of cell viability of SAOS-2 cells incubated in the presence of varying concentrations of gentamicin (*n* = 6) (**a**). Percentage of cell viability of SAOS-2 cells incubated in the presence of varying concentrations of chitosan nanoparticle components, as well as a sample of C6 nanoparticles (*n* = 6) (**b**).

**Figure 8 antibiotics-13-00208-f008:**
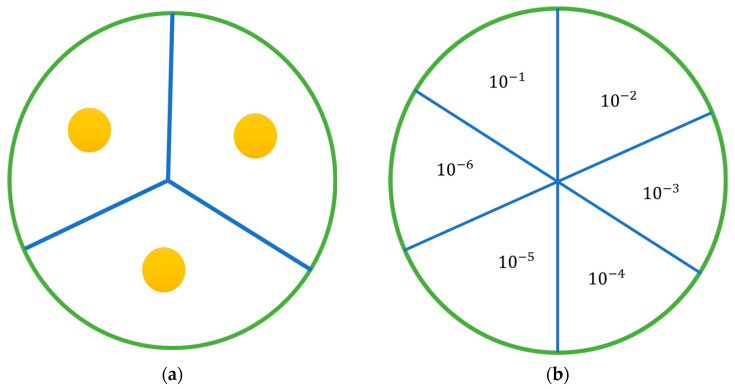
Agar plates: (**a**) diagram of the agar plate with the position of wells; (**b**) diagram of the plates used in the antimicrobial activity study, showing how each plate was divided and the placement of each dilution. Each sample was serially diluted 6 times, and 20 µL from each dilution was plated and incubated.

**Table 1 antibiotics-13-00208-t001:** The most promising formulations from the initial optimization and their respective characteristics (*n =* 6).

Formulation Number	Formulation	Size (nm)	PDI	Zeta Potential (mV)
C4	CS 0.5%: TPP 4%	212.27 ± 19.69	0.33 ± 0.02	−1.80 ± 1.90
C5	CS 1%: TPP 0.1%	236. 08 ± 32.05	0.30 ± 0.3	8.60 ± 2.30
C6	CS 1%: TPP 1%	198.28 ± 9.11	0.32 ± 0.02	4.60 ± 2.57

**Table 2 antibiotics-13-00208-t002:** R^2^ values produced using various release model graphs using the 1% *w*/*v* chitosan and 1% *w*/*v* TPP-loaded formulations with gentamicin.

Formulation Number	R^2^ Zero Order	R^2^ First Order	R^2^ Higuchi Release	Best Fit Model
CG0.5	0.634	0.8055	0.7502	First
CG1	0.8618	0.5188	0.9366	Higuchi
CG2	0.915	0.8189	0.9739	Higuchi

**Table 3 antibiotics-13-00208-t003:** Formulation scheme.

Drug Loading Concentration (% *w*/*v*)	Formulation Number
	Gentamicin
0.5	CG0.5
1	CG1
2	CG2

**Table 4 antibiotics-13-00208-t004:** Concentration and amount of standard plated in 50 µL for standards used in zone of inhibition assays.

Aerobic Strains (*P. aeruginosa* and *S. aureus*)	
Gentamicin Concentration Plated (mg/mL)	Amount of Gentamicin Plated (µg)
0.25	12.5
0.5	25
1	50

## Data Availability

The authors confirm that the data supporting the findings of this study are available within the article.

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
