# Peer review of "Evaluation of the Potential of Chitosan Nanoparticles as a Delivery Vehicle for Gentamicin for the Treatment of Osteomyelitis"

_antibiotics, 2024, doi:10.3390/antibiotics13030208_

Round 1

Reviewer 1 Report

Comments and Suggestions for Authors

The study is well conducted and results are well presented. Overall the paper has merit and clinical applications. However, few more experiments need to be performed to improve the quality of the paper.

Please find my comments below:

1) the zone of inhibition pictures should be included. The agar well/disc diffusion assay..

2) XRD and SEM should be performed for nanoparticles characterization.

3) there is a sudden drug release for few formulations (around 40%) and then sustained release. What is the reason??

4) Also mention if the drug release is fickian or non-fickian type of diffusion.

5) Try not to include values in the discussion section. Some of the sentences are repetition of results.

6) Improve the discussion section by comparing your results with recent literature/references.

7) Recheck the manuscript for spelling, grammer, scientific English. Also check for abbreviations.

Comments on the Quality of English Language

 Recheck the manuscript for spelling, grammer, scientific English. Also check for abbreviations.

Author Response

Antibiotics Editorial Office

30 December 2023

Dear Sir or Madam,

Re: Reviewer comments for ‘Evaluation of the Potential of Chitosan Nanoparticles as a Delivery Vehicle for Gentamicin for the treatment of Osteomyelitis’ for submission to the Special issue – Novel Delivery Systems and Approaches for Antibiotics.

The authors would like to thank the reviewers for their review of the research paper and comments provided, which have been addressed below.

Reviewer 1

The study is well conducted and results are well presented. Overall the paper has merit and clinical applications. However, few more experiments need to be performed to improve the quality of the paper.

Please find my comments below:

  • The zone of inhibition pictures should be included. The agar well/disc diffusion assay.

Images of zones of inhibition do not add to the submission as the resolution is not clear. Figures 4a and 4b display the size of standards and formulations that were carried out in triplicate.

  • XRD and SEM should be performed for nanoparticles characterization.

SEM images have been added in Section 2.6. Morphological analysis of alginate-NPs powders.

  • There is a sudden drug release for few formulations (around 40%) and then sustained release. What is the reason??

The potential reason was included in the discussion however has been made clearer. This is discussed in lines 264 to 271.

  • Also mention if the drug release is fickian or non-fickian type of diffusion.

Drug release characteristcs have been investigated and included in Section 2.4.1.

  • Try not to include values in the discussion section. Some of the sentences are repetition of results.

The discussion section has been reviewed. The team feel that the addition of results draws the reader’s attention to the main discussion points for ease of reference.

  • Improve the discussion section by comparing your results with recent literature/references.

The discussion has been further reviewed and extra literature has been included.

  • Recheck the manuscript for spelling, grammer, scientific English. Also check for abbreviations

The paper was reviewed several times prior to submission however this has again been reviewed by external colleagues.

Many thanks for the opportunity to address any concerns regarding the research paper.

Thank you

Deborah Lowry

Reviewer 2 Report

Comments and Suggestions for Authors

Major issues

- The authors should make the abstract more informative by adding the quantitative information for each results.
- The authors should present the statistical analysis for each result obtained. There is no statistical analysis section in the manuscript.
- The authors should improve the presentation of the results. They did not properly described the results in the text.
- The authors should perform cytotoxic evaluations for the preparation. 
- It is not clear for this review what are the parameters that make these formulation a candidate for the treatment of Osteomyelitis. This relationship should be better explained. 

Author Response

Antibiotics Editorial Office

30 December 2023

Dear Sir or Madam,

Re: Reviewer comments for ‘Evaluation of the Potential of Chitosan Nanoparticles as a Delivery Vehicle for Gentamicin for the treatment of Osteomyelitis’ for submission to the Special issue – Novel Delivery Systems and Approaches for Antibiotics.

The authors would like to thank the reviewers for their review of the research paper and comments provided, which have been addressed below.

Reviewer 2

- The authors should make the abstract more informative by adding the quantitative information for each results.

The abstract has been undated to include quantitative results.

- The authors should present the statistical analysis for each result obtained. There is no statistical analysis section in the manuscript. 

Further statistics has been included throughout the document.

- The authors should improve the presentation of the results. They did not properly described the results in the text.

Presentation of results has been reviewed in the results section.

- The authors should perform cytotoxic evaluations for the preparation. 

Cytotoxic results have been added in Section 2.7 Effect of Gentamicin, Chitosan, TPP and blank formulation on cell viability.

- It is not clear for this review what are the parameters that make these formulation a candidate for the treatment of Osteomyelitis. This relationship should be better explained. 

Desired formulation parameters have been added to the discussion section in lines 203 to 206.

Many thanks for the opportunity to address any concerns regarding the research paper.

Thank you

Deborah Lowry

Reviewer 3 Report

Comments and Suggestions for Authors

 The manuscript reports evaluation of physicochemical properties and antibacterial activity of gentamicin containing chitosan nanoparticles using in vitro assays. I think it is unfinished work and lack of novelty for publication.

 The nanoparticles for the local antibiotic therapy for the treatment of osteomyelitis require good local bioavailability of systemic antibiotics while diminishing possible off-target effects. However, the manuscript just showed in vitro antibacterial activity, there is no experimental data in terms of dosage form, in vivo experiments, etc., which must be included in the manuscript to clarify the potential of the nanoparticles for the treatment of osteomyelitis. There's a lot of work to do.

 As far as I know there are some improved drug delivery systems that include gentamicin and chitosan, although they are not cited in the manuscript. So I don’t understand how significant the nanoparticle is compared to the known delivery systems, and also gentamicin-impregnated PMMA beads, which have been well known for the treatment of osteomyelitis. The following are the list.

   1. Gentamicin-loaded chitosan/folic acid-based carbon quantum dots nanocomposite hydrogel films as potential antimicrobial wound dressing. Journal of Biological Engineering, 2022, 16, 36.

   2. Gentamicin-Ascorbic Acid Encapsulated in Chitosan Nanoparticles Improved In Vitro Antimicrobial Activity and Minimized Cytotoxicity. Antibiotics (Basel), 2022, 11, 1530.

   3. Gentamicin release from chitosan and collagen composites. Journal of Drug Delivery Science and Technology, 2016, 35, 353-359.

   4. Electrophoretic deposition of gentamicin and chitosan into titanium nanotubes to target periprosthetic joint infection. Journal of biomedical materials research, 2023, 111, 1697-1704.

 Brief comment should be added about the results showed in Figure 1 for ease of understanding.

Author Response

Antibiotics Editorial Office

30 December 2023

Dear Sir or Madam,

Re: Reviewer comments for ‘Evaluation of the Potential of Chitosan Nanoparticles as a Delivery Vehicle for Gentamicin for the treatment of Osteomyelitis’ for submission to the Special issue – Novel Delivery Systems and Approaches for Antibiotics.

The authors would like to thank the reviewers for their review of the research paper and comments provided, which have been addressed below.

Reviewer 3

 The manuscript reports evaluation of physicochemical properties and antibacterial activity of gentamicin containing chitosan nanoparticles using in vitro assays. I think it is unfinished work and lack of novelty for publication.

The paper presents the potential of the nanoparticles and represents finished work. This includes pre-formulation, formulation optimization, assay development, drug delivery studies, pharmacokinetics, antimicrobial studies and cell cytotoxicity studies.

 The nanoparticles for the local antibiotic therapy for the treatment of osteomyelitis require good local bioavailability of systemic antibiotics while diminishing possible off-target effects. However, the manuscript just showed in vitro antibacterial activity, there is no experimental data in terms of dosage form, in vivo experiments, etc., which must be included in the manuscript to clarify the potential of the nanoparticles for the treatment of osteomyelitis. There's a lot of work to do.

This is a developmental project to ensure that the nanoparticles are optimized for the application which is referred to in the title and the abstract. The authors ensure that only essential in vivo work is carried out and follow the three R’s.

 As far as I know there are some improved drug delivery systems that include gentamicin and chitosan, although they are not cited in the manuscript. So I don’t understand how significant the nanoparticle is compared to the known delivery systems, and also gentamicin-impregnated PMMA beads, which have been well known for the treatment of osteomyelitis. The following are the list.

  1. Gentamicin-loaded chitosan/folic acid-based carbon quantum dots nanocomposite hydrogel films as potential antimicrobial wound dressing. Journal of Biological Engineering, 2022, 16, 36.
  2. Gentamicin-Ascorbic Acid Encapsulated in Chitosan Nanoparticles Improved In Vitro Antimicrobial Activity and Minimized Cytotoxicity. Antibiotics (Basel), 2022, 11, 1530.
  3. Gentamicin release from chitosan and collagen composites. Journal of Drug Delivery Science and Technology, 2016, 35, 353-359.
  4. Electrophoretic deposition of gentamicin and chitosan into titanium nanotubes to target periprosthetic joint infection. Journal of biomedical materials research, 2023, 111, 1697-1704.

 Further discussion on the controlled release nature of the delivery system has been added. The issue with the PMMA beads is highlighted due to the lack of controlled release. The aim of the current work is to explore controlled release of antimicrobials and hopefully lead to a reduction in antimicrobial resistance.

 Brief comment should be added about the results showed in Figure 1 for ease of understanding.

Further details have been added to Figure 1.

Many thanks for the opportunity to address any concerns regarding the research paper.

Thank you

Deborah Lowry

Round 2

Reviewer 2 Report

Comments and Suggestions for Authors

Dear authors,

I could not find the effects of the suggestions provided during the first review round in the updated manuscript. For instance, I could not see the inclusion of statistical analysis in the results (in tables and figures),

In this sense, I can not accept this manuscript.

Author Response

Dear Sir or Madam,

Re: Reviewer comments for ‘Evaluation of the Potential of Chitosan Nanoparticles as a Delivery Vehicle for Gentamicin for the treatment of Osteomyelitis’ for submission to the Special issue – Novel Delivery Systems and Approaches for Antibiotics.

The authors would like to thank the reviewers for their review of the research paper and comments provided, which have been addressed below.

Reviewer 2 (Round 2)

I could not find the effects of the suggestions provided during the first review round in the updated manuscript. For instance, I could not see the inclusion of statistical analysis in the results (in tables and figures),

The authors apologise for not being clear in the previous draft. All points have been addressed and highlighted in the text.

Statistical analysis have been included in a separate section in 2.8 and added into figure 1a.

The conclusions section has been improved by identifying the optimised formulation based on the results.

Many thanks for the opportunity to address any concerns regarding the research paper.

Thank you

Deborah Lowry

Reviewer 3 Report

Comments and Suggestions for Authors

My previous comments in my report remain unclear due to the authors' responses, which appear to be pointless. However, I would like to leave the final decision to the editor.

Author Response

Dear Sir or Madam,

Re: Reviewer comments for ‘Evaluation of the Potential of Chitosan Nanoparticles as a Delivery Vehicle for Gentamicin for the treatment of Osteomyelitis’ for submission to the Special issue – Novel Delivery Systems and Approaches for Antibiotics.

The authors would like to thank the reviewers for their review of the research paper and comments provided, which have been addressed below.

Reviewer 3 (Round 2)

My previous comments in my report remain unclear due to the authors' responses, which appear to be pointless. However, I would like to leave the final decision to the editor.

The authors are disappointed that the comments did not address the reviewers’ queries and the reviewer finds these ‘pointless’. The team are happy to address any concerns the reviewer has or provide further explanation to the formulation and development of the system.

Many thanks for the opportunity to address any concerns regarding the research paper.

Thank you

Deborah Lowry